# A bioinspired self-powered optical tactile sensing system with ultrahigh sensitivity and ultralow detection limit

Tingting Hou[1,2], Chaojie Chen [1,2], Ru Guo[1,2], Shaoshuai He[2] & Yunlong Zi [1,2,3] ✉

High-sensitivity tactile perception is vital for precise robotic operations in human-machine interactions (HMI). Currently, state-of-the-art tactile perception relies on electrical sensors which demand complex circuits and decoding components, leading to increased energy consumption and susceptibility to electromagnetic interference. Hence, the utilization of human-perceptible signals, such as visible light, as the transmission medium is necessary. Inspired by the mechano-electro-optical transduction mechanism in dinoflagellate bioluminescence, here we propose a self-powered optical tactile sensing system (SOTS) for converting the electrical signal of pressure sensing into visible luminescent intensity, thereby enabling the wireless transmission and visualization of tactile information. The proposed SOTS features an ultrahigh sensitivity (22.4 kPa$^{-1}$) and an ultralow detection limit (10 Pa) in optical tactile sensing with the ultra-wide dynamic range across 5 orders of magnitude (0.01-100 kPa), opening promising avenues for ultrasensitive feedback and intuitive understanding of haptic perception in future HMI.

Tactile perception is the core sense for humans to instantaneously perceive the surroundings, communicate emotions, and ensure safety[1,2]. With advancements in artificial intelligence, numerous sophisticated robots have been equipped with advanced tactile sensors, enabling them to execute accurate operations and facilitate human-like interactions with the environment[3,4]. Currently, many tactile perception systems rely on different types of electrical sensors that require complex circuit designs and decoding components[5-7], suffering from electromagnetic interference (EMI). Besides, the demand for distributed energy supply leads to high costs for power maintenance in the long term. Therefore, incorporating intuitive human-readable signals, such as visible light, as the transmission medium can eliminate electrical wire connections and EMI, thus providing an alternative wireless communication approach.

Mechanoluminescence (ML), capable of swiftly converting mechanical stimuli into a spectrum of visible light, has emerged as a leading technology for intuitive visualized tactile perception[8-14]. However, the threshold pressure of ML is typically at the MPa level[15],

hindering its ability to detect gentle mechanical triggering (e.g., ~Pa) and achieve high sensitivity. Moreover, the low luminous intensity of ML-based optical sensors restricts their functionality in ambient light[16,17], thereby significantly limiting their potential application scenarios. In the realm of visualization platforms, the advent of alternating current electroluminescence (ACEL) devices has also brought revolutionary changes to the field of smart visualized sensing[18-21]. It can emit vivid and bright visible light under alternating current (AC) input for diverse functional applications and interactive scenarios. Despite numerous advantages, ACEL-based technology relies primarily on electrical excitation and lacks tactile feedback as compared to ML-based methods, resulting in limited application in haptic perception. In addition, it still faces issues with post-installation power maintenance due to the high energy consumption of ACEL[21]. Hence, it is essential to develop novel artificial perception approaches that can directly convert subtle mechanical triggers into bright optical signals without external power supply, enabling high-sensitivity tactile sensing for human-machine interactions (HMI).

[1]Department of Mechanical and Automation Engineering, The Chinese University of Hong Kong, Shatin, N.T., Hong Kong, China. [2]Thrust of Sustainable Energy and Environment, The Hong Kong University of Science and Technology (Guangzhou), Guangzhou, Guangdong, China. [3]Division of Integrative Systems and Design, Hong Kong University of Science and Technology, Kowloon, Hong Kong, China. ✉e-mail: ylzi@hkust-gz.edu.cn

Biological systems offer unparalleled inspiration for advancing state-of-the-art bionic electronics[22,23]. In nature, bioluminescent organisms are widely distributed across the marine and terrestrial environments, emitting visible light while being stimulated by various external factors. Among them, bioluminescence of dinoflagellates, which responds to mechanical stress (waves, flows, etc.) for intraspecific communication, predator defense and illumination, has been extensively studied[24]. Remarkably, stress levels as low as several Pa can activate bioluminescence of dinoflagellate[23], demonstrating an extraordinary sensitivity to mechanical stress. Extensive research has also revealed that the mechanism of dinoflagellate bioluminescence primarily involves an intracellular mechano-electro-optical conversion[25], inspiring us to develop an ultrasensitive tactile sensor for visualized artificial perception.

In this work, a biomimetic self-powered optical tactile sensing system (SOTS) is developed that incorporates a custom-built capacitive pressure sensor (CPS) for stress perception and an ACEL unit for real-time visualized sensing. The application of pressure on the sensor induces variations in its capacitance, thereby enabling the modulation of the voltage loaded on ACEL units, which controls the luminous intensity. Distinct pressure magnitudes correspond to distinguishable luminous gradients, enabling visualized quantification of tactile stimuli. Compared with existing visualized sensors, the SOTS achieves an ultrahigh sensitivity (22.4 kPa$^{-1}$) and an ultralow detection limit (10 Pa), along with an ultra-wide detection range spanning five orders of magnitude (0.01–100 kPa), paving the way for advanced visualized tactile sensing. Furthermore, by integrating the SOTS into a robotic hand, real-time visualized responses to tactile stimuli for accurate handling of fragile objects are demonstrated through remote interaction with a real human hand, with the light emission from the SOTS observable even under the ambient light. Therefore, the proposed SOTS is expected not only to promote the sensitivity and enlarge the detection range of visualized tactile perception, but also to expand the scope of applications in HMI.

## Results

### Biomimetic mechanism of the SOTS

Bioluminescence, the emission of visible light by organisms as a result of natural chemical reaction, is a widespread phenomenon in nature[24,26]. In this context, we present the SOTS, a design inspired by stress-induced dinoflagellate bioluminescence, which is a prevalent coastal phenomenon known as "blue tears". As shown in Fig. 1a, dinoflagellates, as unicellular marine organisms[27,28], employ light emission primarily for intraspecific communication and predator deterrence[23]. Light production occurs within specialized organelles termed scintillons, containing luciferin and luciferase. Its bioluminescence is mainly excited by external mechanical stimuli, such as ocean wave-induced stress[25]. At the cellular level, visible light is mediated by a cascade of processes spanning stimulus perception to photon emission. The chemical reaction to produce light is pH-dependent, necessitating the acidification of the scintillons[29]. Initially, the external stress exerted on the cell membrane triggers a mechano-transduction pathway to generate an action potential across the scintillon membrane by increasing cytosolic Ca$^{2+}$ concentration[30], which results in the activation of voltage-gated proton channels, facilitating the influx of H$^+$ into the scintillon and thus decreasing the internal pH[29,31]. Then, the reaction of luciferin and luciferase can be activated under the influence of oxygen, resulting in the production of photons. In summary, the dinoflagellate bioluminescence is mediated by mechano-electro-optical transduction, enabling visible light emission even under gentle wave agitation.

Inspired by the transduction process of dinoflagellates, we engineered SOTS, a sensitive visualized tactile perception system, by integrating a self-developed CPS with an ACEL unit, as demonstrated in Fig. 1b and the photograph of the physical setup is presented in Supplementary Fig. 1. The regulation of ACEL luminance by the CPS is primarily based on capacitance-mediated voltage division effect. Specifically, the CPS exhibits minimal capacitance in the absence of external tactile stimulation, leading to the lowest voltage division across the ACEL unit and thus a notably low initial luminance level. Upon application of tactile stimuli, the capacitance of the CPS increases, resulting in a higher voltage allocation to the ACEL unit and a real-time enhancement of light intensity. At the microscale, the luminescence mechanism of ACEL unit can be attributed to hot-electron impact excitation[18]. The electrons are excited from ground state ($S_O$) to excited state ($S_I$) in the phosphor-dielectric matrix interface under the electric field generated by high-AC voltage, then followed by the photon emission upon radiative de-excitation process from $S_I$ to $S_O$. This is why the SOTS enables the conversion of mechanical stress into bright visible light by biomimetic principles in design, offering a paradigm-shift strategy for visualized tactile perception.

### Design and sensing properties of the CPS

To facilitate a more intuitive understanding of the working principle of SOTS, the equivalent circuit diagram is presented in Fig. 2a. Within this setup, tactile sensors, as a critical component for achieving highly sensitive visualized tactile perception, require mechano-electro transduction with high fidelity to ensure accurate haptic signal acquisition. Therefore, the development of pressure sensors with impedance matched to ACEL and mechanical robustness is essential. Here, we construct a fully soft CPS by sandwiching stretchable silicone (Ecoflex 00-30) as the dielectric layer with PDMS/MWCNTs electrodes and Kapton films serving as the encapsulation layer, as shown in Fig. 2b. The fabrication process is illustrated in Fig. 2c and detailed in Materials and methods. This layer-by-layer curing method is designed to ensure a strong interlayer bond (Fig. 2d and Supplementary Fig. 2), avoiding electrical breakdown and thus contributing to the mechanical stability of the CPS.

Thereafter, the capacitive output of the CPS is accurately quantified with the pressure of up to 100 kPa, exhibiting a nearly linear relationship, as presented in Fig. 2e. Besides, the CPS features great sensing response and excellent recovery stability in a large pressure range (Fig. 2f), laying a solid foundation for tactile sensing. Moreover, a signal-to-noise of 3:1 can be defined as the detection limit of the sensor[32,33]. Here, the threshold pressure of the CPS can reach up to 5 kPa (Fig. 2g) primarily due to the impact of EMI on the electrical measurement, underscoring the necessity of introducing EMI-immune transmission media, such as visible light. To evaluate the dynamic response of the sensor, a 50 kPa load was applied to it, followed by rapid release, revealing a response time of 67 ms and a relaxation time of 80 ms (Supplementary Fig. 13), which are comparable to human skin (~50 ms). Additionally, superior mechanical durability under prolonged or cyclic loading is also critical for flexible pressure sensors to achieve reliable signals. Hence, the repeated compression/release test over 1200 cycles with a peak value of 72 kPa was performed, and the sensor exhibits no sign drift or fluctuation (Fig. 2h) during the cyclic tests, indicating our CPS as a promising candidate for haptic perception.

### Luminescent performance of the ACEL units

Optical components need to achieve high-efficiency electro-optical conversion to enable wireless visual transmission of tactile information, thereby imposing rigorous demands on luminescence intensity and optical differentiability. In this case, a conventional AC electroluminescent light source comprises ZnS-loaded phosphors in soft polymer matrix, sandwiched between flexible transparent electrodes, as illustrated in Fig. 3a. The principle of light emission primarily arises from the hot-electron impact excitation under an AC high-voltage electric field, with the detailed process demonstrated in Fig. 1b. Here, a rotation-mode triboelectric nanogenerator (R-TENG) is employed as the AC power source to provide high-voltage input to the ACEL unit (Supplementary Fig. 6), thereby exciting bright visible light. Because of triboelectric effect and electrostatic induction, the R-TENG can

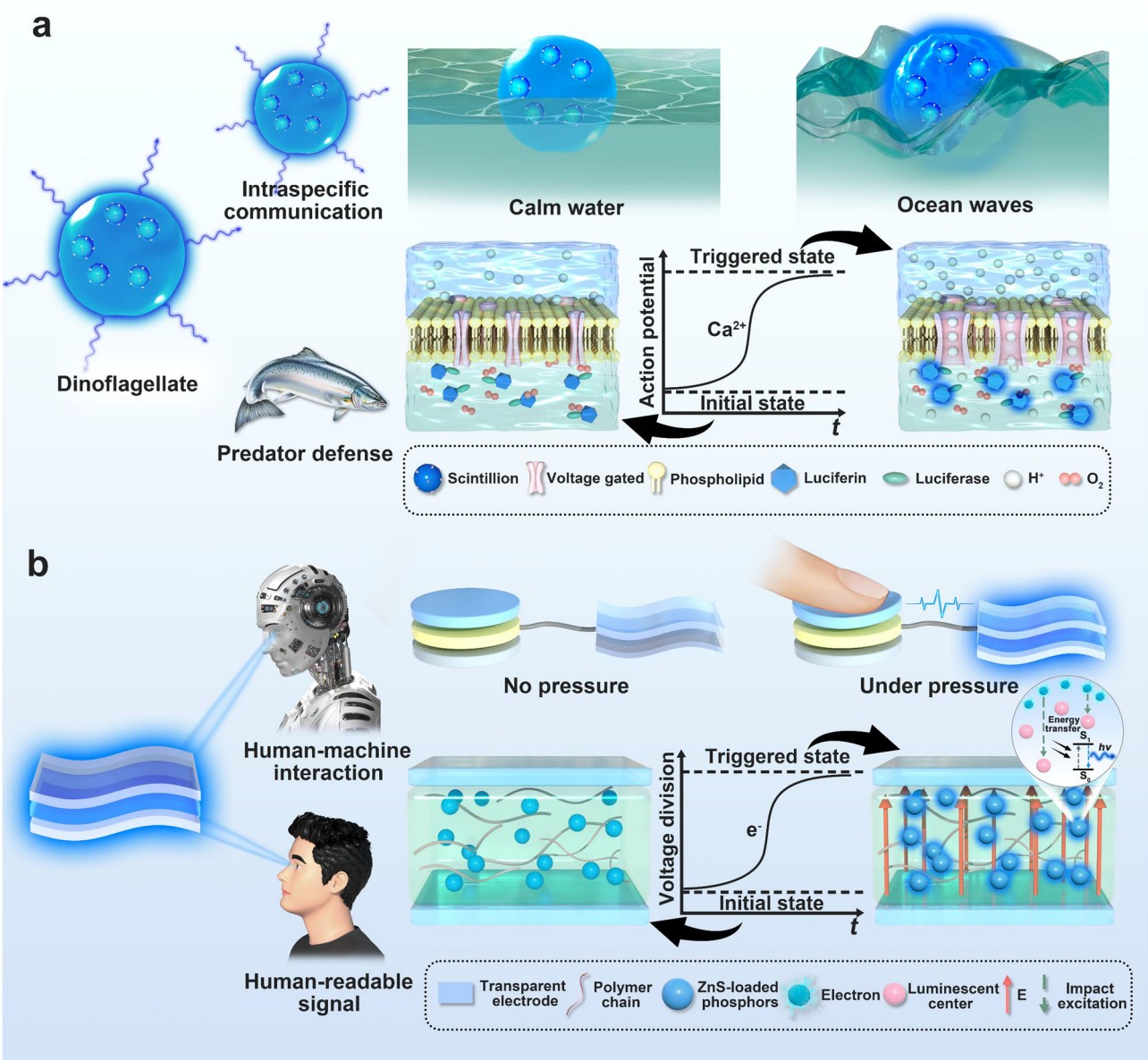

**Fig. 1 | Schematic illustration of the biomimetic SOTS. a** The illustration explains the functions of dinoflagellate bioluminescence and the principles of light emission under the mechanical action of wave impact. **b** The schematic illustrates the applications of SOTS and the principles of its biomimetic design.

effectively convert mechanical energy into electricity without external power sources[34,35]. The structural configuration, working principle and voltage outputs of the R-TENG are depicted in Supplementary Fig. 3. To further demonstrate the self-powered capability of the R-TENG, a handle was mounted on the rotor to facilitate manual operation, allowing it to generate high-voltage outputs through hand cranking, as illustrated in the Supplementary Fig. 8.

Next, the luminescent performance of self-powered ACEL units for visualized sensing was systematically evaluated by adjusting the energy inputs and device parameters. Figure 3b and Supplementary Fig. 7 show the effects of applied voltage and frequency on the luminous energy density and brightness of the ACEL unit, revealing a monotonic enhancement in both indicators with increasing voltage and frequency. The relationship between luminescence (L) and applied voltage (V) at a fixed frequency is given by ref. 36

$$L = L_0 \exp\left(-\beta/V^{0.5}\right) \tag{1}$$

where $L_0$ and $\beta$ are both device/material-dependent constants. Correspondingly, the spectra of luminous power under different frequencies and voltages are clearly shown in Fig. 3c, demonstrating a pronounced improvement trend with a fixed emission center of 515 nm within the spectral range of 400–700 nm. Besides, a series of ACEL units with parametrically modulated active areas and dielectric layer thickness were engineered and investigated to establish a foundational framework for advancing visualized sensing. The corresponding capacitances of ACEL units with different sizes have been summarized in Supplementary Table 1. As shown in Fig. 3d, it is observed that the light intensity of ACEL units decreases with increasing active area under a constant frequency. This is because the ACEL unit can be simply modeled as a parallel-plate capacitor, of which the capacitance ($C_E$) can be expressed as:

$$C_E = \frac{\varepsilon_r \varepsilon_0 A}{d} \tag{2}$$

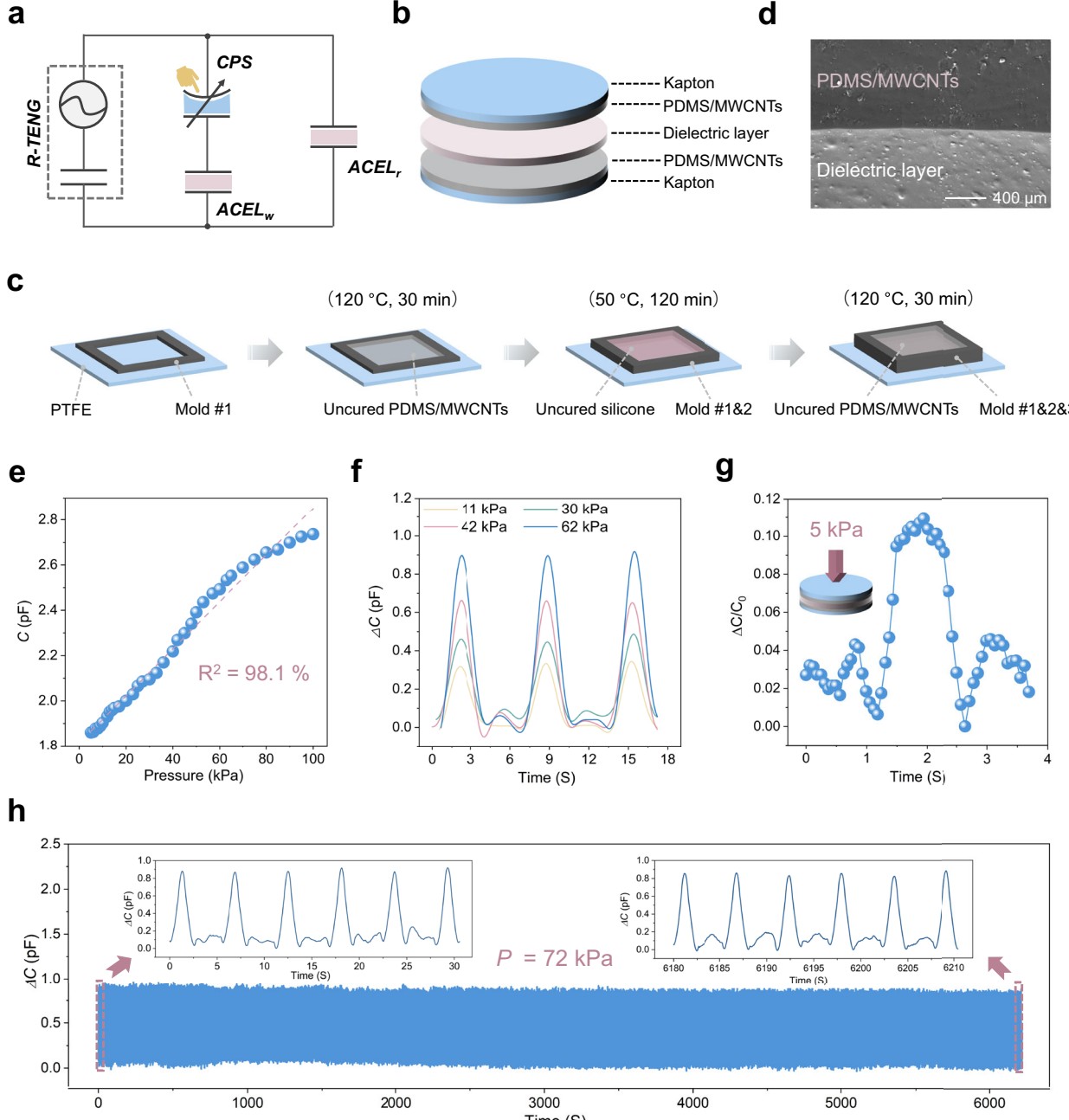

**Fig. 2 | The structure and characterizations of the CPS. a** Equivalent circuit diagram of the SOTS ($ACEL_w$ denotes the working ACEL unit, while $ACEL_r$ refers to the reference ACEL unit). **b** Schematic of the structure and composition of the CPS. **c** The fabrication process of the CPS to achieve a strong interlayer bond. **d** SEM image of the interface between the dielectric layer and the PDMS/MWCNTs electrode, demonstrating the close connection between them. **e** Variation of capacitance over the pressure range up to 100 kPa. **f** Sensing response and recovery stability under different pressures. **g** Limit of the detection. **h** Working stability tested over 1200 cycles under 72 kPa.

where $\varepsilon_r$ and $\varepsilon_0$ is the relative and vacuum permittivity respectively, and $A$ refers to the active area and here $d$ is the thickness of the dielectric layer. Additionally, the total open-circuit voltage output ($V_{OC}$) and voltage division across the components are as follows:

$$Voc = V_T + V_E = \frac{Q}{C_T} + \frac{Q}{C_E} \qquad (3)$$

Where $V_T$ and $V_E$ stand for the voltage divisions for R-TENG and ACEL unit, respectively, and $C_T$ is the internal capacitance of R-TENG, which is usually considered as a constant. When the dielectric thickness remains constant (200 μm), the increased active areas (2 × 2, 3 × 3, 4 × 4 cm²) results in a higher $C_E$ (Eq. 2), leading to a reduction in its voltage division $V_E$ according to Eq. 3 and thus a corresponding decrease in luminescence due to Eq. 1, as well as the luminous energy density (Supplementary Fig. 4b) and luminous power (Supplementary Fig. 4a). The corresponding visualized images of ACEL units powered by the R-TENG with constant frequency and open-circuit voltage (4.5 kV) are presented in Supplementary Fig. 4c, illustrating a significant variation in brightness. The luminescence of ACEL units with varied dielectric layer thickness (300, 200, 100 μm) and the same the active area (2 × 2 cm²) is shown in Fig. 3e. As powered by the R-

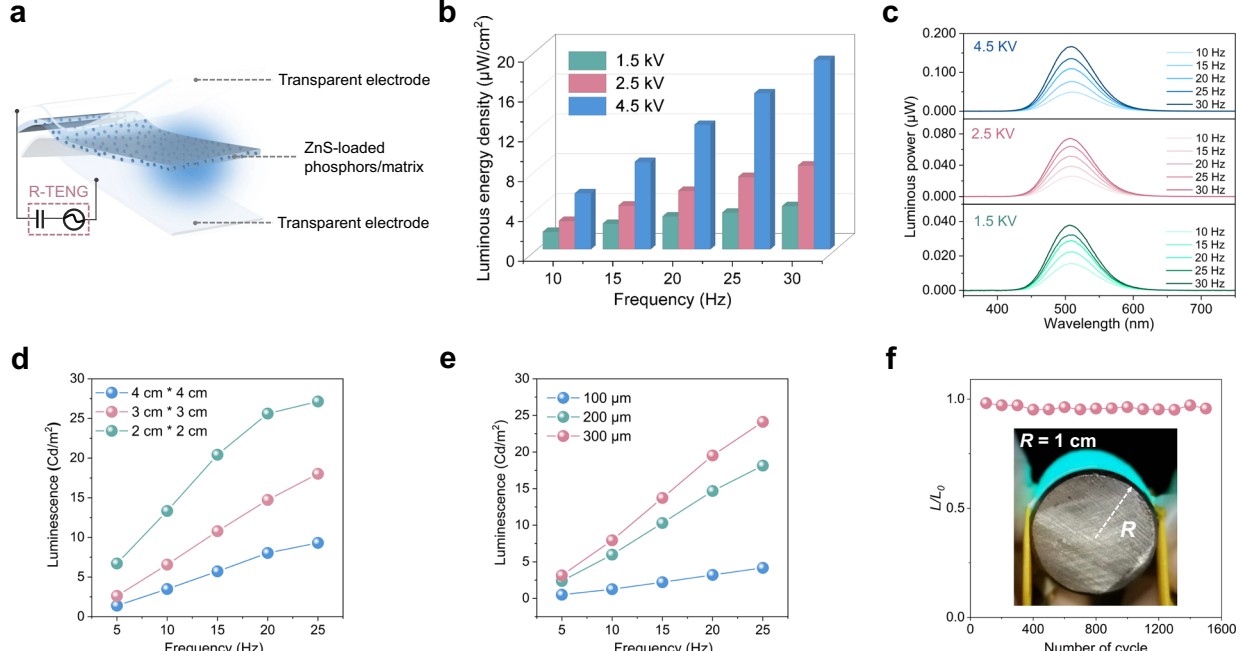

**Fig. 3 | The structure and performance of the ACEL units. a** Schematic of the flexible ACEL unit, which comprises ZnS-loaded phosphors embedded in stretchable silicone as the luminescent emission layer and PET/ITO as the transparent electrodes. **b** Effects of voltage on the luminous energy density of ACEL unit at different frequency inputs (area: 3 cm × 3 cm, thickness: 200 μm). **c** The spectra of luminous power under different applied voltage and frequencies (area: 3 cm × 3 cm, thickness: 200 μm). **d** The luminescence of ACEL units with varied active areas under different frequencies (thickness: 200 μm, 4.5 kV-powered). **e** The luminescence of ACEL units with varied dielectric thickness under different frequencies (area: 3 cm × 3 cm, 4.5 kV-powered). **f** Light emission stability test of the flexible ACEL units during bending-recovery operations and the inset shows the degree of bending (R stands for radius of curvature).

TENG, a noticeable attenuation in luminescence is observed with the decreasing thickness, which originates from the increased capacitance $C_E$ of the ACEL unit due to Eq. 2 and hence the decreased voltage division $V_E$ due to Eq. 3, thereby lowering the brightness (Supplementary Fig. 5c) according to Eq. 1 as well as the luminous power, luminous energy density (Supplementary Fig. 5a, b). This behavior is contrary to the typical trend observed in ACEL devices powered by a conventional AC source, where luminescence increases with reduced thickness owing to the monotonic dependence of electric field on thickness. Accordingly, a detailed investigation was conducted to examine the dependence of the ACEL device's luminance on its thickness, revealing a critical trade-off point (Supplementary Fig. 9).

To ensure the flexibility and mechanical stability of the optical components, the ACEL unit was subjected to cyclic bending tests under continuous electrical excitation by R-TENG. After 1500 bending-recovery cycles, the luminescence ($L$) of unit retained >95% of its initial luminance ($L_O$), which is even visible under ambient light, as exhibited in Fig. 3f. This exceptional electromechanical robustness and optoelectronic stability confirm our ACEL units as superb candidates toward wireless visualized interactive media.

### Optical tactile responsive performance of SOTS

The marriage of CPS and ACEL units enables effective mechano-electro-optical transduction, facilitating the intuitive wireless visualized transmission of haptic information. To achieve high sensitivity and low detection limit for gentle tactile feedback, ACEL units with varying structural parameters are coupled with CPS to identify the optimal configuration. Here, we define the luminance sensitivity as $S = \triangle L/\triangle P$, in which $\triangle L$ and $\triangle P$ represent the variation of luminescence intensity and applied pressure, respectively. As shown in Fig. 4a, although the 2 × 2 cm² unit demonstrates slightly higher luminance sensitivity than that of the 3 × 3 cm² unit within 1 kPa, with the identical thickness of 200 μm, its high baseline of luminescence without loading due to the increased voltage division reduces the signal-to-noise ratio

for the visualized perception of subtle tactile signals. Besides, while keeping the area of 3 × 3 cm², the ACEL unit with a 200 μm dielectric thickness exhibits the highest luminescent sensitivity (0.52 cd/m²/kPa), with near-zero baseline without loading (Fig. 4b). Therefore, ACEL units with 3 × 3 cm² active area and 200 μm dielectric thickness were selected as the optimal parameters for the optical components in SOTS, to optimize the tactile sensitivity. All subsequent experimental protocols employed these configurations unless otherwise specified. The SOTS exhibits an unprecedented high sensitivity over a wide pressure range up to 100 kPa (Fig. 4c). To facilitate the comparison between our study and existing previous studies, we adopted the relative luminescent sensitivity, which is defined as $S_R = \triangle L/L_O/\triangle P$, while $L_O$ is the initial luminescence intensity without pressure loading. The average relative luminescent sensitivity is $S_{R1} \sim 22.4\ \text{kPa}^{-1}$ when the pressure is below 1 kPa and $S_{R2} \sim 1.16\ \text{kPa}^{-1}$ for pressure >1 kPa with 10 Hz input. The corresponding luminescence characteristics of ACEL units under varied pressure and frequency inputs are systematically presented in Fig. 4d and the detailed luminance behavior within this pressure range is illustrated in Supplementary Fig. 10, demonstrating exceptional and intuitive optical differentiability.

More remarkably, the SOTS achieves a record-low detection limit of 10 Pa via optical signal as the transmission media, as shown in Fig. 4e, which is 500 times lower than that via sole CPS by electrical signals (Fig. 2g), due to the superior EMI-immunity of visible-light-mediated signal communication over traditional electrical paradigms. Furthermore, as summarized in Fig. 4f, the SOTS shows an incomparably high sensitivity, ultra-low limit of detection and an ultrabroad work range of pressure across 5 orders of magnitude, to the best of our knowledge, outperforming existing optical tactile sensing devices in other reported literatures[37–47]. It is also worth noting that the sensitivity of SOTS within 1 kPa (22.4 kPa⁻¹) exceeds the highest reported values (2.6 kPa⁻¹) in existing work by 8.3 times, demonstrating its superior capability in sensing subtle tactile stimuli as compared to conventional devices. To further illustrate the exceptional sensitivity and ultra-low

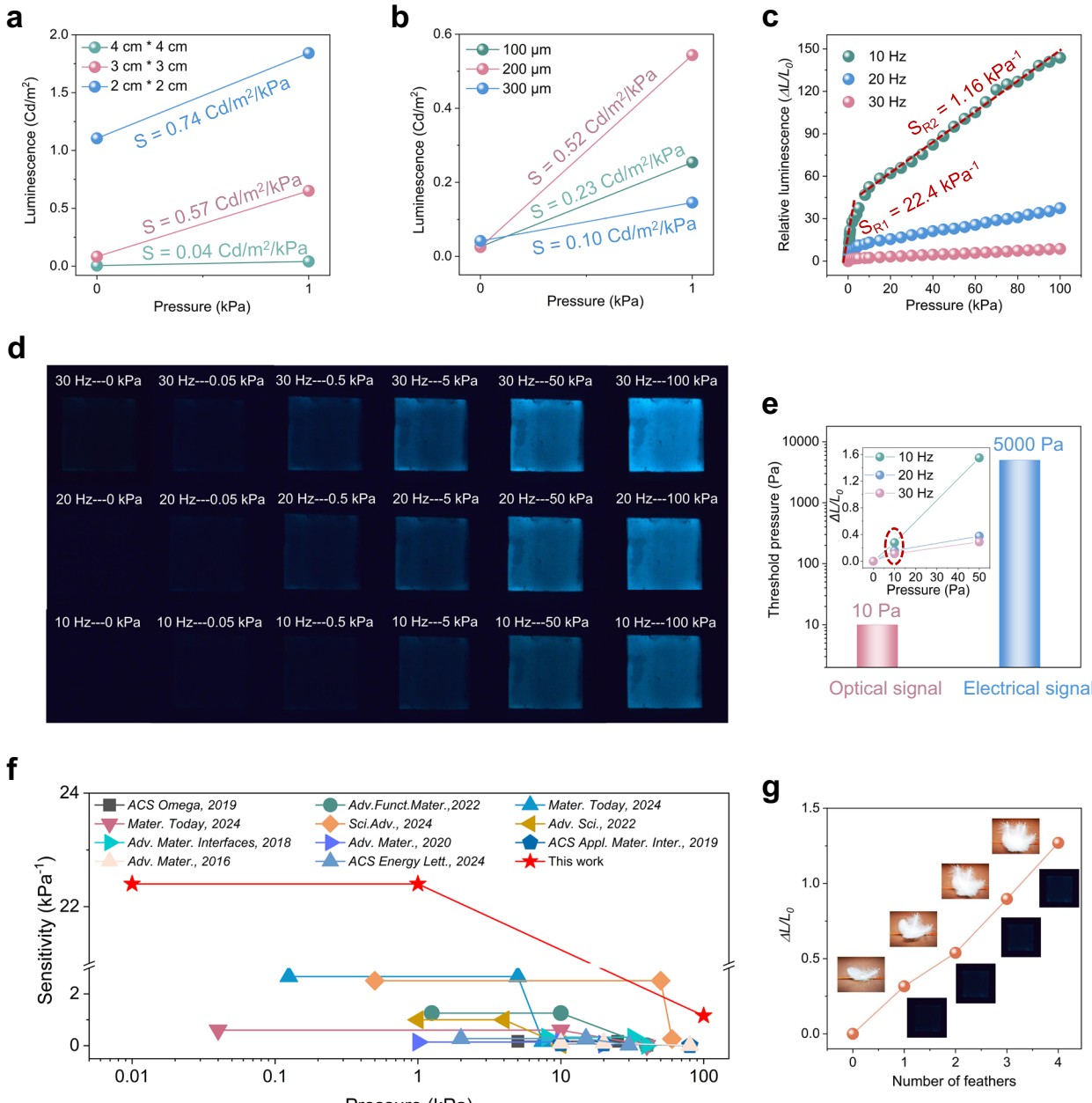

**Fig. 4 | Optical tactile sensing performance of SOTS. a** The optical response sensitivity of ACEL units with different active areas at pressures ranging from 0 to 1 kPa (dielectric thickness: 200 μm). **b** The optical response sensitivity of ACEL units with different dielectric thickness at pressures ranging from 0 to 1 kPa (active areas: 3 cm × 3 cm). **c** The luminescence sensitivity of SOTS with the pressure up to 100 kPa under varying frequency inputs. **d** A visual representation of the ACEL units under pressures ranging from 0 to 100 kPa, along with varying frequency inputs.

**e** The comparison of threshold pressure for optical signals from SOTS and electrical signals only from CPS (the inset indicates that the detection limit of SOTS is 10 Pa). **f** Comparison of the sensitivity, detection limit and sensing range of proposed SOTS with previous reported optical tactile sensing devices. **g** The optical response to varying quantities of feathers on the CPS, illustrating the ultra-low detection limit and exceptional sensitivity of SOTS.

detection limit of SOTS, here, different quantities of feathers were placed on the CPS, and the corresponding optical response were captured, as illustrated in Fig. 4g. Therefore, the proposed SOTS is promising for advancement of visualized subtle tactile perception with excellent sensing performance.

**Visualized subtle tactile perception enabled by SOTS**
To validate the feasibility of SOTS in enabling subtle tactile perception in HMI for robots, we engineered the CPS unit into a robotic hand for intact grasping and manipulation of fragile objects (e.g., eggs), with teleoperated control implemented through human-guided hand

motions (Fig. 5a). This system comprises a somatosensory-interaction glove equipped with a Bluetooth module for wireless communication, a robotic hand for replicating the movements of the human hand, and a CCD camera for capturing and transmitting optical signals. When the human hand remotely controls the robotic hand to grasp fragile objects, the lack of subtle tactile sensing hinders the user's ability to assess the loading pressure, increasing the risk of damaging the item (Fig. 5b). So, the integration of high-sensitivity pressure sensors on robotic hand is necessary, as it enables real-time haptic feedback to users, thereby facilitating immediate loading corrections. To ensure the fidelity of the optical signal from SOTS, a reference ACEL unit was

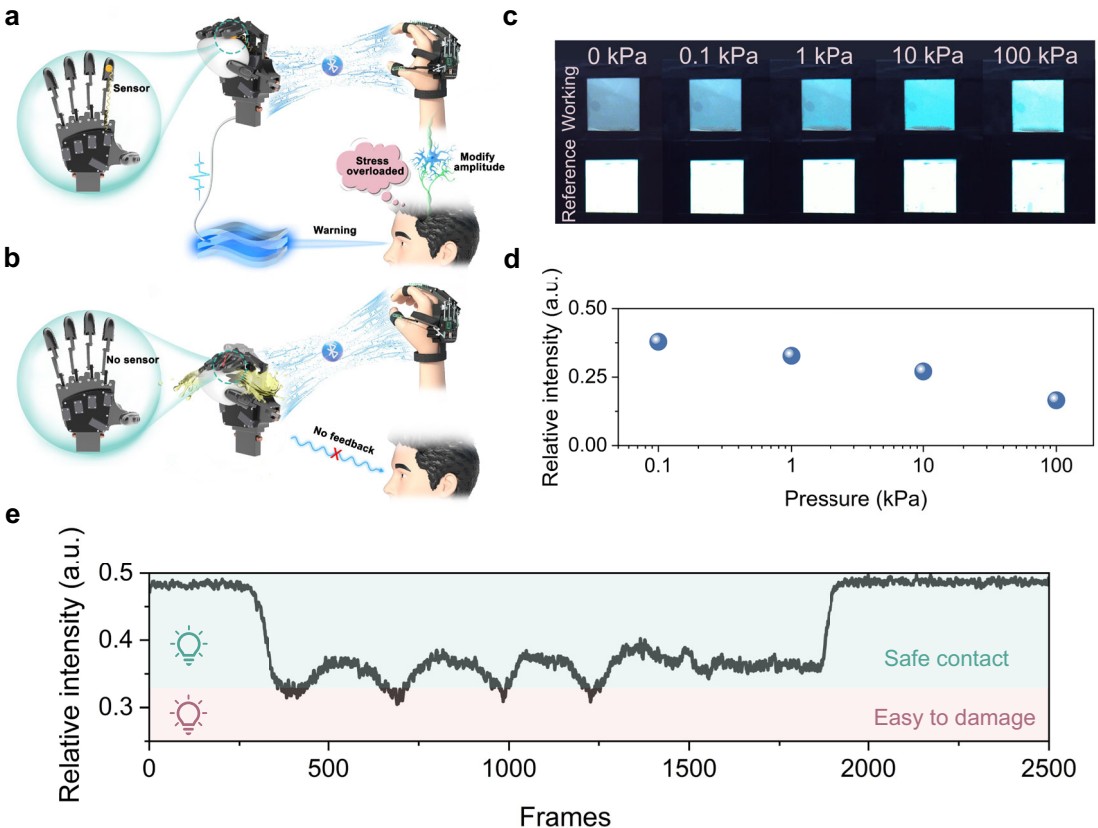

**Fig. 5 | The visualized precise tactile perception enabled by SOTS.** Practical application scenarios for handling fragile objects with SOTS (**a**) and without SOTS (**b**) in the context of remote HMIs. The luminance variations (**c**) and their relative intensity profiles (**d**) of the working and reference ACEL units under varied loaded pressures in ambient light. **e** Variations of relative light intensity can provide users with intuitive visualized feedback, enabling them to promptly correct the magnitude of the applied pressure.

integrated into the system for real-time calibration of luminescence drift during dynamic tactile sensing operations (Fig. 2a). The luminance outputs from both the reference and working ACEL units were captured by CCD camera and processed via a custom-developed algorithm to quantify their relative intensity for intuitive tactile pressure sensing. Besides, the luminance variations of the working and reference ACEL units in ambient light under different loaded pressures, along with their relative intensity profiles, are presented in Fig. 5c, d, demonstrating visually discernible luminescence intensities that correlate with the applied tactile stimuli. Additionally, Supplementary Fig. 11 shows the luminance variations of the working and reference ACEL units captured in the darkroom, confirming the robust luminescent distinguishability of the SOTS across all tested environmental conditions. Hence, through the effective integration of SOTS into mechanical fingers, a robotic hand with ultrasensitive tactile perception is realized, allowing accurate and secure grasping of fragile objects during remote interactions with human users (Supplementary Movie 2 and Fig. 5e).

## Discussion

Inspired by the mechano-electro-optical transduction mechanism in dinoflagellate bioluminescence, we proposed a paradigm-shift strategy for ultra-sensitive visualized tactile sensing through SOTS, as integrated from CPS as the tactile perception unit, R-TENG as the power source, and an ACEL device as the light source. The engineered CPS alters the luminescence of light-emitting units in response to external mechanical stimulation via the capacitance-modulated voltage division, facilitating biomimetic visualized response of tactile perception even in daylight. Through a detailed investigation of the

optimal synergistic performance of CPS and ACEL unit, SOTS has demonstrated the highest relative luminescence sensitivity (22.4 kPa$^{-1}$) and the lowest detection limit (10 Pa) over an ultrawide pressure range across 5 orders of magnitude (0.01–100 kPa). Based on these exceptional metrics, a robotic hand equipped with our SOTS was developed for intactly grasping fragile objects through remote HMI. Therefore, this ultrasensitive SOTS holds the potential for advancing high-precision somatosensory interaction and provides insights for developing the next-generation visualized tactile sensing technologies.

## Methods

### Materials

ZnS: Cu phosphors (D502CT) were purchased from Shanghai KPT Co., Ltd. Multi-walled carbon nanotubes (MWCNTs) (diameter: 10–20 nm; length: 2–8 μm) were purchased from Shenzhen Yuechuang Technology Co., Ltd. The PDMS silicone elastomer kit (Sylgard 184) was purchased from Dow chemical company. Ecoflex (00-30) was purchased from Smooth-On, Inc. PET/ITO (0.125 mm-thick) and PTFE template (area: 5 cm × 5 cm, thickness: 3 mm) are commercially available. All materials were used as received without further treatment.

### Preparation of PDMS/MWCNTs precursor

MWCNTs were sonicated with the PDMS precursor and an appropriate amount of ethanol for 60 min. The mixture was then heated in an oven at 60 °C until the ethanol was completely evaporated. Subsequently, a PDMS curing agent was added in a 1:10 ratio with precursor, and sonication continued for an additional 30 min to ensure uniform mixing.

## Fabrication of the CPS

A smooth template (PTFE) and a high-temperature-resistant mode #1 are well prepared, followed by uncured PDMS/MWCNTs pouring into the mode and then heated at an elevated temperature (120 °C) for 30 min as bottom electrode (0.2 mm in thickness). Next, the Ecoflex precursor (00-30, A/B = 1/1) is directly poured onto the cured bottom electrode and constrained in terms of dimensions (1.2 cm × 1.2 cm) and thickness (2.4 mm) via mold #2, then curing at 50 °C for 120 min. Finally, another PDMS/MWCNTs precursor is also directly cured on the surface of the middle dielectric layer as the top electrode, using the same preparation conditions as those for the bottom electrode. This layer-by-layer curing method is to achieve structural design with tight interlayer connections, preventing electrical breakdown. Besides, the CPS is easily scaled by adjusting the size (thickness and area) while maintaining other parameters are the same.

## Fabrication of ACEL units

The ZnS: Cu phosphors were uniformly mixed with Ecoflex (00-30, A/B = 1/1) in a 2:1 ratio. The mixture was then evenly coated onto commercially available PET/ITO flexible transparent electrodes with different diameters. After allowing the coated substrate to dry at room temperature for 20 min, a second transparent electrode was placed on top. This sandwich structure was then cured in an oven at 50 °C for 2 h.

## Characterization and measurement

The SEM image was captured by using a high-resolution scanning electron microscope (SU5000). The loaded pressure was evaluated by force gauges (IMADA ZTS-5N, YLICE DS2-100N), and the capacitance was measured by using an LCR meter (TH2830). Periodic mechanical motion was performed by a linear motor (LinMot H01) and electrical output of R-TENG was tested and recorded by an electrostatic meter (Keithley 6514). The luminescent performance was recorded simultaneously by a commercialized system (XPQY-EQE-Adv, Guangzhou Xi Pu Optoelectronics Technology Co., Ltd.) that was equipped with an integrated sphere and photodetector array, and the corresponding optical images were captured via an industrial camera.

## Data availability

Source data are provided with this paper.

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

## Acknowledgements

This work was funded by National Key R&D Program of China (2024YFB3816000), Guangdong Natural Science Funds for Distinguished Young Scholar (Grant no. 2023B1515020074), and Guangzhou-HKUST(GZ) Joint Funding Project (Grant no. 2024A03J0466). We acknowledge the facility and service supports from the Materials Characterization and Preparation Facility (GZ) (MCPF (GZ)) at the Hong Kong University of Science and Technology (Guangzhou). We acknowledge the support from Wilson Tang Brilliant Energy Science and Technology Lab (BEST Lab) at the Hong Kong University of Science and Technology (Guangzhou).

## Author contributions

Y.Z. and T.H. conceived the idea. T.H. performed all experiments with assistance from C.C., R.G., and S.H. in code development, sensor fabrication, and SEM imaging, respectively. T.H. conducted data analysis and visualization under the supervision of Y.Z. T.H. drafted the original manuscript, and both Y.Z. and T.H. contributed to its revision. Y.Z. was responsible for funding acquisition.

## Competing interests

The authors declare no competing interests.
