## [Transparent Peer Review file · Nature Communications]

A bioinspired self-powered optical tactile sensing system with ultrahigh sensitivity and ultralow detection limit

Corresponding Author: Professor Yunlong Zi

Version 0:

Reviewer comments:

Reviewer #1

(Remarks to the Author)

This manuscript reports a novel optical tactile sensor with high sensitivity and low detection limit. The tactile sensor was constructed by three components, including a capacitive pressure sensor (CPS) as a tactile perception unit, a rotation-mode triboelectric nanogenerator (R-TENG) as a power source, and an alternating current electroluminescence (ACEL) device as a light source. This design enabled the tactile sensor to response to external mechanical stimulus at an extremely low level. I have noticed some issues, which I believe should be addressed before it could be considered for publication on Nature Communications.

1. The authors emphasized that the constructed optical tactile sensor as a self-powered sensor. Actually, the optical tactile sensor could not work without the R-TENG as a power source. Although the R-TENG was an unit to convert mechanical actions into high-voltage electricity, the high-frequency rotation to drive the R-TENG did not come from a applied stress (low-frequency), and I believe that the source driving the R-TENG to work in this work was electricity. As shown in Fig. S4, the role of R-TENG was exactly equivalent to a traditional AC-power. In this case, I think it is unreasonable to define it as a self-powered tactile sensor.
2. The use of R-TENG as the power source may generate other conceptional confusion. The R-TENG converts high-frequency mechanical actions to electricity and further excite phosphors to emission photons. This seems to be a mechano-to-photon conversion process. However, applying a stress on the tactile sensor to generate “great enhancement” of photon emission should not be attributed to mechano-to-photon conversion. It is just due to the variation of the capacitance of sensor.
3. The device structure of the tactile sensor is not clearly presented in the manuscript. The authors show depict the connection between the CPS, R-TENG, and ACEL (e.g. in Fig. 1B), that can demonstrate the working principle of the sensor without unnecessary confusion.
4. Figures S4-S6 in Supporting Information are not mentioned in the manuscript.

Reviewer #2

(Remarks to the Author)

This manuscript presents a bio-inspired self-powered optical tactile sensor (SOTS) that integrates a capacitive pressure sensor (CPS), an alternating current electroluminescence (ACEL) unit, and a rotation-mode triboelectric nanogenerator (R-TENG) as a power source. The authors claim a record-high sensitivity of 22.4 kPa^{-1} and a detection limit of 10 Pa, enabling real-time visualized tactile sensing for applications such as robotic manipulation. However, several technical and conceptual issues remain unresolved. In particular, the self-powered claim is not convincingly demonstrated, and the visibility and practical usability of the system under realistic conditions are questionable. This manuscript has insufficient novelty to be published in this journal, and many aspects require significant improvement.

1. The manuscript claims a “self-powered” system, yet the R-TENG is driven by a linear motor (LinMot H01). To justify this claim, the authors should provide data demonstrating the ability of the R-TENG to operate under naturally available mechanical inputs, such as hand motion, environmental vibration, or human interaction, without reliance on powered actuation.
2. The manuscript lacks detailed information regarding the size, weight, and form factor of the R-TENG. Since the generator is intended for integration into robotic hands, information on its mechanical compatibility, flexibility, and size constraints is critical.
3. The relative sensitivity is highly sensitive to the baseline luminance. In optical tactile sensors, the baseline luminance

often approaches zero. This may lead to overestimated sensitivity values. Thus, it remains unclear how all sensitivity values were calculated consistently across studies in Figure 4F. To fairly support your performance claims, provide a comparative table summarizing the baseline luminance and absolute sensitivity values of your device and references.

4. The connection between dinoflagellate bioluminescence and the sensor mechanism appears somewhat indirect. The device relies on capacitance change and voltage division, which fundamentally differs from biological processes. The use of "bio-inspired" should be more carefully justified.

5. Figures 3B/C, 3E/F, and 3G/H present overlapping data using different units (luminance vs. luminous energy density).

6. The author explains the relationship between the size of the ACEL device and its luminance in terms of capacitance.

However, to support this theory, capacitance should be measured for devices of different sizes.

7. In a typical ACEL device, an increase in thickness leads to a decrease in the electric field, resulting in lower luminance.

However, since this device is serially connected with a R-TENG, the influence of capacitance also contributes to an increase in luminance with thicker devices. It would be helpful to show the trade-off point, where the decrease in luminance due to reduced electric field and the increase due to enhanced capacitance balance out, as a function of thickness.

8. Although the Supporting Information suggests the components are connected in series using circuit diagram, no clear image or schematic is provided showing the full system.

9. The authors claim that the system operates effectively under ambient or daylight conditions. However, it appears that the luminance of integrated device in Figure 4C may be lower than that of single ACEL unit (Figure 3) even though the luminance of the single ACEL device (~20 cd/m²) may be insufficient under typical daylight conditions. Furthermore, all the luminescence images provided under darkroom conditions. The authors must provide actual luminance of integrated device and experimental evidence supporting visibility under realistic daylight conditions.

10. The definition and calculation of "relative intensity" shown in Figures 5D and 5E remain unclear. Moreover, the "custom-developed algorithm" used for processing the optical signals is not described at all.

11. The authors report a detection limit of 10 Pa based on luminescence output. However, the generated luminescence may be insufficient for practical visibility, particularly under ambient lighting. The authors should clarify how this detection limit was determined and whether it corresponds to meaningful tactile perception in realistic environments.

12. The Materials and Methods section fails to specify whether the materials were synthesized or purchased commercially.

Reviewer #3

(Remarks to the Author)

Optical transmission is a key channel in human-machine interaction, and tactile sensing is essential for enabling a robot human-like perception. The manuscript reported a self-powered optical tactile sensor inspired by dinoflagellate bioluminescence for real-time visualized tactile perception. By utilizing a self-fabricated pressure sensor as the tactile recognition unit, an electroluminescent component for optical emission and a triboelectric nanogenerator for power supply, the tactile stimuli can be transduced into corresponding optical intensity in a self-powered way. Their device also demonstrated record-high luminescent sensitivity, record-low detection limit in mechano-optical conversion with ultrawide working range. In addition, a robotic hand with high-sensitivity visual tactile recognition capabilities has been demonstrated, highlighting the promising application prospects of their sensor. Therefore, I highly recommend this article for publication in Nature Communications once the following questions are addressed:

Q1: The sensor's response time to pressure is also important for assessing its mechanical performance. Could the authors provide a test to evaluate the dynamic response speed of the capacitive pressure sensor?

Q2: In the section on "Design and sensing properties of the CPS", it is mentioned that a signal-to-noise ratio of 3:1 can be defined as the detection limit of the sensor. Please provide a reference to support this statement.

Q3: Please provide a CIE chromaticity diagram of the emitted light.

Q4: In Fig. 5C, the authors have demonstrated the capability for visual tactile sensing under ambient light through relative brightness between a working ACEL and a reference ACEL. Could the authors also provide a visual comparison under dark conditions?

Q5: In Fig. S2, the output voltage of R-TENG is tested by introducing a 5 GΩ resistor. Could the author also provide the corresponding output current?

Version 1:

Reviewer comments:

Reviewer #1

(Remarks to the Author)

I have read the updated manuscript and the response letter. I find that the authors have revised the manuscript properly according to the reviewers' comments. I believe that the manuscript in the present form is ready for publication.

Reviewer #2

(Remarks to the Author)

Authors addressed all the comments raised by the reviewers

Reviewer #3

(Remarks to the Author)

Response to reviewers for the manuscript (NCOMMS-25-38951-T)

We would like to thank all the reviewers for taking your valuable time to read our manuscript thoroughly and give your in-depth advice on our work. Their constructive comments and concerns have been carefully addressed and answered in a point-by-point manner. For the convenience of the reviewers, the comments and suggestions are listed below in **blue font**, followed by our responses in normal **black font**. We also highlighted the corresponding modifications in our revised manuscript in **red font**. We believe that the questions proposed by the reviewers have significantly improved the quality of our manuscript, making it more suitable for publication in *Nature Communications*.

Reviewer #1:

This manuscript reports a novel optical tactile sensor with high sensitivity and low detection limit. The tactile sensor was constructed by three components, including a capacitive pressure sensor (CPS) as a tactile perception unit, a rotation-mode triboelectric nanogenerator (R-TENG) as a power source, and an alternating current electroluminescence (ACEL) device as a light source. This design enabled the tactile sensor to response to external mechanical stimulus at an extremely low level. I have noticed some issues, which I believe should be addressed before it could be considered for publication on Nature Communications.

Response: We thank the reviewer for your kind comments and willingness to provide valuable advice on our work.

1. The authors emphasized that the constructed optical tactile sensor as a self-powered sensor. Actually, the optical tactile sensor could not work without the R-TENG as a power source. Although the R-TENG was an unit to convert mechanical actions into high-voltage electricity, the high-frequency rotation to drive the R-TENG did not come

from a applied stress (low-frequency), and I believe that the source driving the R-TENG to work in this work was electricity. As shown in Fig. S4, the role of R-TENG was exactly equivalent to a traditional AC-power. In this case, I think it is unreasonable to define it as a self-powered tactile sensor.

Response: We appreciate the constructive comments provided by the reviewer. Indeed, The R-TENG serves as the power source within the entire circuit, converting mechanical motions into electrical energy based on the contact electrification and electrostatic induction. We have carefully reviewed the established concept of “self-powered sensor” in the literature (e.g., *Chem. Rev.*, 2023, 123(21): 12105-12134; *Energ. Environ. Sci.*, 2015, 8(8): 2250-2282). As defined in these key references, this term primarily applies when the nanogenerator itself functions as the sensing unit without requiring an external power supply. Conversely, when a nanogenerator powers other functional electronic components (such as the ACEL and CPS elements in our system), the configuration is more accurately termed a “self-powered system”. Therefore, in line with this established distinction and specific role of the R-TENG in powering both the ACEL and CPS devices, we consider “self-powered optical tactile sensing system” to be a more accurate and appropriate expression of our work, which has been consistently adopted throughout the revised manuscript.

Modification: Title, Page 1

A biomimetic self-powered optical tactile **sensing system** with record-high sensitivity and record-low detection limit

Modification: Line 19-24, Page 2

Inspired by the mechano-electro-optical transduction mechanism in dinoflagellate bioluminescence, here we propose a self-powered optical tactile **sensing system** (SOTS) based on a custom-built capacitive pressure sensor and an alternating current electroluminescence unit for real-time visualized tactile perception, by converting the

electrical signal of pressure sensing into visible luminescent intensity.

Modification: Line 80-82, Page 4

In this work, a biomimetic self-powered optical tactile **sensing system** (SOTS) is developed that incorporates a custom-built capacitive pressure sensor for stress perception and an ACEL unit for real-time visualized sensing.

2. The use of R-TENG as the power source may generate other conceptional confusion. The R-TENG converts high-frequency mechanical actions to electricity and further excite phosphors to emission photons. This seems to be a mechano-to-photon conversion process. However, applying a stress on the tactile sensor to generate “great enhancement” of photon emission should not be attributed to mechano-to-photon conversion. It is just due to the variation of the capacitance of sensor.

Response: We sincerely appreciate the reviewer’s suggestion. To avoid this conceptual confusion, the term “mechano-optical conversion” has been revised to “optical tactile sensing” in the manuscript.

Modification: Line 26-28, Page 2

To date, the proposed SOTS features a record-high sensitivity (22.4 kPa^{-1}) and a record-low detection limit (10 Pa) in **optical tactile sensing** with the ultra-wide dynamic range across 5 orders of magnitude (0.01-100 kPa).

Modification: Line 86-90, Page 4

Compared with existing visualized sensors, the SOTS achieves a record-high sensitivity (22.4 kPa^{-1}) and a record-low detection limit (10 Pa) ~~in mechano-optical transduction,~~ **along** with **an** ultra-wide detection range **spanning** 5 orders of magnitude (0.01-100 kPa), setting new benchmarks for visualized tactile sensing techniques.

3. The device structure of the tactile sensor is not clearly presented in the manuscript. The authors show depict the connection between the CPS, R-TENG, and ACEL (e.g. in Fig. 1B), that can demonstrate the working principle of the sensor without unnecessary confusion.

Response: We greatly appreciate the reviewer's suggestion. To clearly illustrate the interconnections between the various components (CPS, R-TENG and ACEL) within the self-powered optical tactile sensing system, we have supplemented the equivalent circuit diagram (Fig. 2a) and a physical photograph, as shown in Supplementary Fig. 1 (Page 2, Supplementary Materials). The CPS and ACEL can be simply modeled as parallel-plate capacitors. These two units are connected in series within the circuit, with the R-TENG serving as the power source. The regulation of ACEL luminance by the CPS is primarily based on a capacitance-mediated voltage division mechanism. Specifically, the CPS exhibits minimal capacitance in the absence of external tactile stimulation, leading to the lowest voltage division across the ACEL unit and thus its minimum initial luminance. Upon application of tactile stimuli, the capacitance of the CPS increases, resulting in a higher voltage allocation to the ACEL element and a corresponding enhancement in luminous intensity. As a result, the pressure of the tactile stimuli can be directly converted into the magnitude of optical intensity, enabling the visualization and wireless transmission of tactile information. Besides, to mitigate environmental influences and ensure the fidelity of the optical signal, a reference ACEL unit ($ACEL_r$) was involved in the SOTS system to enable real-time calibration of potential luminance deviations during dynamic tactile sensing.

Modification: Line 117-126, Page 5-6

Inspired by the transduction process of dinoflagellates, we engineered SOTS, a sensitive visualized tactile perception system, by integrating a self-developed capacitive pressure sensor (CPS) with an ACEL unit, as demonstrated in Fig. 1b and the photograph of the physical setup is presented in Supplementary Fig. 1. The

regulation of ACEL luminance by the CPS is primarily based on capacitance-mediated voltage division effect. Specifically, the CPS exhibits minimal capacitance in the absence of external tactile stimulation, leading to the lowest voltage division across the ACEL unit and thus a notably low initial luminance level. Upon application of tactile stimuli, the capacitance of the CPS increases, resulting in a higher voltage allocation to the ACEL unit and a real-time enhancement of light intensity.

Fig. 2. The structure and characterizations of the CPS. **a** Equivalent circuit diagram of the SOTS ($ACEL_w$ denotes the working ACEL unit, while $ACEL_r$ refers to the reference ACEL unit). **b** Schematic of the structure and composition of the CPS. **c** The

fabrication process of the CPS to achieve a strong interlayer bond. **d** SEM image of the interface between the dielectric layer and the PDMS/MWCNTs electrode, demonstrating the close connection between them. **e** Variation of capacitance over the pressure range up to 100 kPa. **f** Sensing response and recovery stability under different pressures. **g** Limit of the detection. **h** Working stability tested over 1200 cycles under 72 kPa.

Supplementary Fig. 1. The physical photograph of the interconnections among the SOTS.

4. Figures S4-S6 in Supporting Information are not mentioned in the manuscript.

Response: We are grateful to the reviewer for pointing this out. In response to Comment 3, a new Supplementary Fig. 1 has been added to the Supplementary Materials. Consequently, the original Supplementary Fig. 4–6 (previously labeled as Fig. S4-S6) have been renumbered as Supplementary Fig. 5–7 and involved in the manuscript.

Modification: Line 232-236, Page 11-12

As powered by the R-TENG, a noticeable attenuation in luminescence is observed with the decreasing thickness, which originates from the increased capacitance C_E of the ACEL unit due to Eq. 2 and hence the decreased voltage division V_E due to Eq. 3,

thereby lowering the brightness (Supplementary Fig. 5c) according to Eq. 1 as well as the luminous power, luminous energy density (Supplementary Fig. 5a and 5b).

Modification: Line 188-190, Page 10

Here, a rotation-mode triboelectric nanogenerator (R-TENG) is employed as the AC power source to provide high-voltage input to the ACEL unit (Supplementary Fig. 6), thereby exciting bright visible light.

Modification: Line 198-201, Page 10

Fig. 3b and Supplementary Fig. 7 show the effects of applied voltage and frequency on the luminous energy density and brightness of the ACEL unit, revealing a monotonic enhancement in both indicators with increasing voltage and frequency.

Reviewer #2:

This manuscript presents a bio-inspired self-powered optical tactile sensor (SOTS) that integrates a capacitive pressure sensor (CPS), an alternating current electroluminescence (ACEL) unit, and a rotation-mode triboelectric nanogenerator (R-TENG) as a power source. The authors claim a record-high sensitivity of 22.4 kPa^{-1} and a detection limit of 10 Pa, enabling real-time visualized tactile sensing for applications such as robotic manipulation. However, several technical and conceptual issues remain unresolved. In particular, the self-powered claim is not convincingly demonstrated, and the visibility and practical usability of the system under realistic conditions are questionable. This manuscript has insufficient novelty to be published in this journal, and many aspects require significant improvement.

Response: We would like to express our sincere gratitude to the reviewer for your constructive suggestions, which have been invaluable in strengthening this manuscript. The novelty of this work lies in the proposed optical tactile sensing system inspired by

the dinoflagellate bioluminescence process. This system directly converts the intensity of tactile stimuli into a corresponding human-recognizable visible light gradient, thereby enabling the wireless transmission and visualization of tactile information. Thus, it eliminates the need for specific decoding components associated with conventional electrical tactile sensors and alleviates their inherent electromagnetic interference issues. Furthermore, our proposed SOTS demonstrates record-high outstanding performance in sensitivity, detection limit, and operational range, representing a paradigm-shift design.

1. The manuscript claims a “self-powered” system, yet the R-TENG is driven by a linear motor (LinMot H01). To justify this claim, the authors should provide data demonstrating the ability of the R-TENG to operate under naturally available mechanical inputs, such as hand motion, environmental vibration, or human interaction, without reliance on powered actuation.

Response: Thanks for the suggestion. In this study, the rotational motion of the R-TENG was driven by an electric motor to ensure precise control frequency and facilitate the exploration of experimental laws. In response to the reviewer’s suggestion, the R-TENG was carefully modified by integrating a manually operated handle onto its rotor, as illustrated in added Supplementary Fig. 8 (Page 9, Supplementary Materials). This handle enables manual rotation of the rotor, thereby generating corresponding electrical output and the detailed operation can be found in Response video 1. This R-TENG can be also driven by nature sources such as wind or water flow, as demonstrated previously (*ACS Energy Lett.*, 2021, 6(6): 2343-2350; *Water Res.*, 2024, 252: 121185).

Supplementary Fig. 8. The hand-cranked R-TENG. **a** The photograph of the R-TENG prototype with an integrated handle on the rotor. **b** The measurement principle of voltage output of R-TENG. **c** The current output. **d** The voltage output.

Modification: Line 194-196, Page 10

To further demonstrate the self-powered capability of the R-TENG, a handle was mounted on the rotor to facilitate manual operation, allowing it to generate high-voltage outputs through hand cranking, as illustrated in the Supplementary Fig. 8.

2. The manuscript lacks detailed information regarding the size, weight, and form factor of the R-TENG. Since the generator is intended for integration into robotic hands, information on its mechanical compatibility, flexibility, and size constraints is critical.

Response: We are very sorry for causing such a misunderstanding to the reviewer. The R-TENG, a disk with a diameter of 28 cm and a mass of 812 g as shown in Supplementary Fig. 3a and Supplementary Fig. 1, converts mechanical energy into electrical energy to provide a high AC-voltage input for the ACEL unit. Currently, only

tactile recognition unit (CPS) has been integrated into the robotic hand. This R-TENG is just for demonstrating the proof-of-concept optical tactile sensing system, and it can be either integrated into the robotic hand or serve as a separate power source. In fact, in TENG family, there are varieties of members, e.g. wind-driven TENG, which are compatible, flexible, and small in size to be suitable for integration in robotic systems. And these TENGs can be broadly adopted to drive optical tactile sensing as stated in this manuscript.

Modification: Line 322-325, Page 16

To validate the feasibility of SOTS in enabling subtle tactile perception in HMIs for robots, we engineered the CPS unit into a robotic hand for intact grasping and manipulation of fragile objects (e.g., eggs), with teleoperated control implemented through human-guided hand motions (Fig. 5a).

3. The relative sensitivity is highly sensitive to the baseline luminance. In optical tactile sensors, the baseline luminance often approaches zero. This may lead to overestimated sensitivity values. Thus, it remains unclear how all sensitivity values were calculated consistently across studies in Figure 4F. To fairly support your performance claims, provide a comparative table summarizing the baseline luminance and absolute sensitivity values of your device and references.

Response: Thanks for reviewer's comment. To address the concerns of the reviewer, we have summarized the baseline luminance (L_0) and absolute sensitivity values ($S_A = \Delta L / \Delta P$) of our device and those from previously reported studies, as shown in Response Table 1. In our work, the baseline luminance can be modulated by the energy inputs from the R-TENG, while the high sensitivity of our device is attributed to the well-designed mechano-electro-optical conversion mechanism. Specifically, the applied tactile pressure alters the capacitance of the CPS, which in turn modifies the voltage division across the ACEL unit, ultimately modulating the optical intensity. Therefore, the intermediate capacitance-modulated voltage distribution mechanism

enables the conversion of subtle tactile pressure variations into significant luminance changes, which distinguishes our design from previous approaches.

As summarized in Response Table 1, our device exhibits the highest relative sensitivity in terms of numerical value. However, the comparison on the absolute sensitivity and baseline luminance cannot be done, since almost all of previous studies characterize optical properties using relative optical intensity (a.u.), which makes comparisons across various works impossible. The only previous work using absolute intensities is *Sci. Adv.*, 2024, 10(44), eadq8989, which is also from our group. We believe the unified measurement standard and absolute luminance intensity can facilitate the comparison between various studies, thereby promoting the standardization and continuous progress of this field.

Besides, the authors believe that the use of relative luminescent sensitivity (S_R) could provide a unified framework by mitigating the influence of varying baseline luminance, thereby enabling consistent and fair comparisons across all systems in the standard unit of kPa^{-1} . In our work, this S_R is absolutely not affected by baseline luminance, as the baseline luminance and the working luminance are both proportional to the electrical output from R-TENG in our design. The S_R is only affected by the division of voltage apply between the CPS and ACEL, as determined by the capacitance variation of CPS under pressure. Besides, we also added the reference ACEL (ACEL_r) to eliminate the impact of the variation of R-TENG output.

References	Baseline luminance (L_0)	Low-pressure Absolute Sensitivity (S_{A1})	High-pressure Absolute Sensitivity (S_{A2})	Low-pressure Relative Sensitivity (S_{R1})	Low-pressure Relative Sensitivity (S_{R2})
ACS Omega , 2019, 4(24): 20470-20475.	0.18 a.u.	0.0305 a.u./kPa	0.014 a.u./kPa	0.17 kPa ⁻¹	0.018 kPa ⁻¹
Adv.Funct.Mater. ,2022, 32(26): 2201292.	0.05 a.u.	0.0629 a.u./kPa	0.01 a.u./kPa	1.26 kPa ⁻¹	0.017 kPa ⁻¹
Mater. Today , 2024, 78:10-19.	0.05 a.u.	0.13 a.u./kPa	0.12 a.u./kPa	2.67 kPa ⁻¹	0.17 kPa ⁻¹
Mater. Today , 2024, 79:73-85.	0.1 a.u.	0.06 a.u./kPa	0.01 a.u./kPa	0.602 kPa ⁻¹	0.014 kPa ⁻¹
Sci. Adv. , 2024, 10(44): eadq8989.	0.0004 mW/cm ²	0.001 mW/cm ² /kPa	0.006 mW/cm ² / kPa	2.51 kPa ⁻¹	0.26 kPa ⁻¹
Adv. Sci. , 2022, 9(32): 2203510.	0.2 a.u.	0.2 a.u./kPa	0.03 a.u./kPa	1 kPa ⁻¹	0.042 kPa ⁻¹
Adv. Mater. Interfaces , 2018, 5(4): 1701063.	0.1 a.u.	0.033 a.u./kPa	0.013 a.u./kPa	0.33 kPa ⁻¹	0.014 kPa ⁻¹
Adv. Mater. , 2020, 32(1): 1904988.	0.2 a.u.	0.028 a.u./kPa	0.025 a.u./kPa	0.139 kPa ⁻¹	0.056 kPa ⁻¹
ACS Appl. Mater. Inter. , 2019, 11(14): 13796-13802.	0.4 a.u.	0.03 a.u./kPa	0.005 a.u./kPa	0.075 kPa ⁻¹	0.007 kPa ⁻¹
Adv. Mater. , 2016, 28(31): 6656-6664.	0.3 a.u.	0.03 a.u./kPa	0.007 a.u./kPa	0.1 kPa ⁻¹	0.011 kPa ⁻¹
ACS Energy Lett. , 2024, 9(5): 2231-2239.	0.15 a.u.	0.042 a.u./kPa	0.02 a.u./kPa	0.28 kPa ⁻¹	0.029 kPa ⁻¹
This work	0.02408 cd/m ²	0.5302 cd/m ² /kPa	0.03 cd/m ² /kPa	22.4 kPa ⁻¹	1.16 kPa ⁻¹

Response Table 1. Comparison of the baseline luminance, absolute luminescent sensitivity and relative luminescent sensitivity between our device and those reported in the literature.

4. The connection between dinoflagellate bioluminescence and the sensor mechanism appears somewhat indirect. The device relies on capacitance change and voltage division, which fundamentally differs from biological processes. The use of “bio-inspired” should be more carefully justified.

Response: We thank the reviewer for this comment. Our design primarily mimics the integrated energy transduction pathway of dinoflagellate bioluminescence. Specifically, ocean-wave induced stress (*mechanical*) triggers the generation of an action potential (*electrical*) across the membrane of the light-emitting organelles, followed by activating voltage-gated biochemical reaction responsible for photon emission (*optical*). This elucidates the comprehensive process by which dinoflagellates are capable of emitting visible light in response to gentle external stimuli. In our design, tactile pressure (*mechanical*) modifies the voltage division (*electrical*) across the ACEL unit by altering the capacitance of the CPS. This triggers voltage-mediated hot-electron impact excitation, thereby inducing photon emission (*optical*). This is why our device enables the conversion of tactile stimulation into bright visible light. In essence, dinoflagellate bioluminescence and our design undergo similar mechano-electro-optical transduction process in principle. Therefore, the authors believe it is reasonable to refer to our device as a biomimetic design.

5. Figures 3B/C, 3E/F, and 3G/H present overlapping data using different units (luminance vs. luminous energy density).

Response: Thanks for the suggestion. The original Figs. 3B, 3F, and 3H have been moved to Supplementary Fig. 4, Supplementary Fig. 5 and Supplementary Fig. 7 (Supplementary Materials), and a new Fig. 3 has been assembled.

Fig. 3. The structure and performance of the ACEL units. **a** Schematic of the flexible ACEL unit, which comprises ZnS-loaded phosphors embedded in stretchable silicone as the luminescent emission layer and PET/ITO as the transparent electrodes. **b** Effects of voltage on the luminous energy density of ACEL unit at different frequency inputs (area: 3 cm × 3 cm, thickness: 200 μm). **c** The spectra of luminous power under different applied voltage and frequencies (area: 3 cm × 3 cm, thickness: 200 μm). **d** The luminescence of ACEL units with varied active areas under different frequencies (thickness: 200 μm, 4.5 kV-powered). **e** The luminescence of ACEL units with varied dielectric thickness under different frequencies (area: 3 cm × 3 cm, 4.5 kV-powered). **f** Light emission stability test of the flexible ACEL units during bending-recovery operations and the inset shows the degree of bending (R stands for radius of curvature).

Supplementary Fig. 4. The optical performance of ACEL units with varied active areas under different frequencies. a Luminous power. b luminous energy density, and c visual images.

Supplementary Fig. 5. The optical performance of ACEL units with varied dielectric thickness under different frequencies. a Luminous power. **b** luminous energy density, and **c** visual images.

Supplementary Fig. 7. The optical performance of ACEL units with varied voltage under different frequencies. a Luminescence. **b** visual images.

6. The author explains the relationship between the size of the ACEL device and its luminance in terms of capacitance. However, to support this theory, capacitance should be measured for devices of different sizes.

Response: A great point! Thanks for the suggestion, the capacitance of ACEL units with different sizes were determined using an LCR meter (Cp-Rp mode, 100 kHz) and summarized in Supplementary Table 1 (Page 16, Supplementary Materials).

Active area (cm ²)	Dielectric thickness (μm)	Capacitance (pF)
2×2	200	238
3×3	200	437
4×4	200	717
3×3	100	832
3×3	300	278

Supplementary Table 1. The capacitance of ACEL units with different sizes.

Modification: Line 210-211, Page 11

The corresponding capacitances of ACEL units with different sizes have been summarized in Supplementary Table 1.

7. In a typical ACEL device, an increase in thickness leads to a decrease in the electric field, resulting in lower luminance. However, since this device is serially connected with a R-TENG, the influence of capacitance also contributes to an increase in luminance with thicker devices. It would be helpful to show the trade-off point, where the decrease in luminance due to reduced electric field and the increase due to enhanced capacitance balance out, as a function of thickness.

Response: The author would like to acknowledge the insightful suggestion provided by the reviewer. As the ACEL device is connected with the R-TENG in the circuit (Supplementary Fig. 6a), an increase in the dielectric thickness (d) reduces its capacitance (Eq. 2), thereby leading to a higher voltage drop (V_{ACEL}) across the ACEL device (Eq. 3). Since the luminance of the ACEL is primarily determined by the electric field ($E_{ACEL} = V_{ACEL}/d$), the resultant change in E_{ACEL} , when both V_{ACEL} and d increase, depends critically on the extent of their respective changes. Therefore, to identify the trade-off point where the luminance reduction due to decreased E_{ACEL} resulting from increased d is compensated by the luminance enhancement arising from the higher V_{ACEL} , we carefully fabricated a series of ACEL devices with well-controlled variations in thickness (100-1000 μm) and systematically characterized their corresponding luminescence, as shown in Supplementary Fig. 9 (Page 10, Supplementary Materials). Our experimental results identify 400 μm as the critical thickness-balanced point. That is, when $d < 400 \mu\text{m}$, the enhancement in luminance is primarily governed by the increased V_{ACEL} resulting from its reduced capacitance. Conversely, when $d > 400 \mu\text{m}$, the reduction in luminance due to the increased thickness becomes dominant.

Modification: Line 236-241, Page 12

This behavior is contrary to the typical trend observed in ACEL devices powered by a conventional AC source, where luminescence increases with reduced thickness owing to the monotonic dependence of electric field on thickness. Accordingly, a detailed investigation was conducted to examine the dependence of the ACEL device's luminance on its thickness, revealing a critical trade-off point (Supplementary Fig. 9).

Supplementary Fig. 9. The ACEL device, connected in series with the R-TENG, exhibits luminance variations as a function of its thickness.

The luminance of the ACEL device is primarily determined by the electric field, as defined by the relation $E_{ACEL} = V_{ACEL}/d$. When the ACEL device is connected in series with the R-TENG, an increase in the dielectric thickness (d) also leads to a rise in voltage drop (V_{ACEL}). Therefore, the change in the E_{ACEL} primarily depends on the relative magnitudes of these variations. The results indicate the presence of an inflection point at a thickness of 400 μm . When $d < 400 \mu\text{m}$, the enhancement in luminance is primarily governed by the increased V_{ACEL} resulting from its reduced capacitance, which leads to a net increase in E_{ACEL} . Conversely, when $d > 400 \mu\text{m}$, the reduction in E_{ACEL} due to the increased thickness becomes the dominant factor, resulting in a decrease in luminance.

8. Although the Supporting Information suggests the components are connected in series using circuit diagram, no clear image or schematic is provided showing the full system.

Response: Thanks for the suggestion. We have added a photograph of the physical setup to provide a clearer representation of our entire system, as shown in Supplementary Fig. 1 (Page 2, Supplementary Materials).

Supplementary Fig. 1. The physical photograph of the interconnections among the SOTS.

9. The authors claim that the system operates effectively under ambient or daylight conditions. However, it appears that the luminance of integrated device in Figure 4C may be lower than that of single ACCEL unit (Figure 3) even though the luminance of the single ACCEL device (~ 20 cd/m²) may be insufficient under typical daylight conditions. Furthermore, all the luminescence images provided under darkroom conditions. The authors must provide actual luminance of integrated device and experimental evidence supporting visibility under realistic daylight conditions.

Response: The luminance of a single ACCEL device is discernible in daylight, as evidenced by the inset in Fig. 3f. Due to the voltage division of CPS in the circuit, the integrated device exhibits lower luminance compared to an isolated ACCEL device without voltage division, and the corresponding actual luminance are provided in the Supplementary Fig. 10 (Page 11, Supplementary Materials). Nevertheless, the light emission still remains visible under ambient lighting conditions. It can be noted that the luminescent images presented in Fig. 5c were captured under daylight. To facilitate subsequent programmatic luminance analysis based on RGB values, the peripheral area of the luminescent region was intentionally set to black to minimize interference from other unrelated colors. We sincerely apologize for any confusion caused to the reviewer and greatly appreciate this constructive feedback, which will help prevent potential misinterpretation by readers in the future. For a more straightforward visual comparison,

the corresponding luminescent images obtained under darkroom conditions at the identical applied pressures are also provided, as shown in Supplementary Fig. 11 (Page 12, Supplementary Materials). Under no applied pressure (0 kPa), the low initial capacitance of the CPS leads to dim ACEL emission that is insufficient for capture in darkroom. However, the intrinsic emission region itself of the ACEL unit can remain detectable under ambient light (Fig. 5c). Therefore, the visualization of luminescence and the distinguishability of its optical intensity under ambient light can be demonstrated by Fig. 5c, indicating the feasibility of wirelessly transmitting tactile stimuli via optical signals.

Modification: Line 285-289, Page 14

The corresponding luminescence characteristics of ACEL units under varied pressure and frequency inputs are systematically presented in Fig. 4d and the detailed luminance behavior within this pressure range is illustrated in Supplementary Fig. 10, demonstrating exceptional and intuitive optical differentiability.

Modification: Line 342-345, Page 16-17

Additionally, Supplementary Fig. 11 shows the luminance variations of the working and reference ACEL units captured in the darkroom, confirming the robust luminescent distinguishability of the SOTS across all tested environmental conditions.

Supplementary Fig. 10. The actual luminance of the ACEL unit within SOTS.

Supplementary Fig. 11. The luminance variations **a** and their relative intensity profiles **b** of the working and reference ACEL units under varied loaded pressures in darkroom.

10. The definition and calculation of "relative intensity" shown in Figures 5D and 5E remain unclear. Moreover, the "custom-developed algorithm" used for processing the optical signals is not described at all.

Response: We thank the reviewer for this suggestion. A more detailed description regarding this part has been added to the manuscript.

Modification: Page 12-13, Supplementary Materials

Here, the relative intensity (ΔL) is defined as:

$$\Delta L = \frac{L_R - L_W}{L_R}$$

where L_R and L_W are the light intensities of reference ACEL and working ACEL, respectively. The light intensity of ACEL is captured by CCD camera and extracted the sum of pixel values via Matlab code.

The custom-developed algorithm is used to monitor real-time tactile-optical conversion processes. Upon capturing optical videos of the working and reference ACEL units with a CCD camera, the images are promptly transmitted to a laptop. The self-developed algorithm is then employed to delineate the luminous regions of both ACEL units and calculate the respective sums of their luminous pixels (L_R and L_W) based on RGB values. Hence, the relative light intensity can be obtained followed by execution of the relevant judgment procedure. Specifically, a threshold of relative intensity (L_{TH}) was established, corresponding to the level of tactile pressure that may compromise the structural integrity of grasped object. Therefore, when $\Delta L \geq L_{TH}$, it indicates that the contact between the robotic hand and the grasped object remains within a safe range. Conversely, if $\Delta L < L_{TH}$, it suggests that the force applied by the robotic hand poses a risk of damaging the object, thus necessitating an immediate reduction in the applied pressure. The specific process flow of the algorithm is detailed in the flowchart below:

Supplementary Fig. 12. Process flow of the custom-developed algorithm for monitoring real-time tactile-optical conversion.

11. The authors report a detection limit of 10 Pa based on luminescence output. However, the generated luminescence may be insufficient for practical visibility, particularly under ambient lighting. The authors should clarify how this detection limit was determined and whether it corresponds to meaningful tactile perception in realistic environments.

Response: We greatly appreciate this insightful comment from reviewer. The detection

limit was determined by the minimum pressure that the spectrometer could detect which induces a measurable change in luminescence, as illustrated in the inset of Fig. 4e. Due to the exceptionally low detection limit, the corresponding variations in light intensity are relatively minor. Although visible light serves as a human-perceivable signal, perceptual sensitivity to such subtle optical changes may vary among individuals. Therefore, the authors believe that direct measurements of luminescence using a spectrometer will yield more objective and accurate results, which is meaningful in tactile perception applications in realistic environments.

12. The Materials and Methods section fails to specify whether the materials were synthesized or purchased commercially.

Response: We thank the reviewer for pointing this out. A dedicated "Materials" section has been added to provide a detailed description of the materials employed during the experimental procedures.

Modification: Line 376-383, Page 19

Materials

ZnS: Cu phosphors (D502CT) were purchased from Shanghai KPT Co., Ltd. Multi-walled carbon nanotubes (diameter: 10-20 nm; length: 2-8 μm) were purchased from Shenzhen Yuechuang Technology Co., Ltd. The PDMS silicone elastomer kit (Sylgard 184) was purchased from Dow chemical company. Ecoflex (00-30) was purchased from Smooth-On, Inc. PET/ITO (0.125 mm-thick) and PTFE template (area: 5 cm \times 5 cm, thickness: 3 mm) are commercially available. All materials were used as received without further treatment.

Reviewer #3:

Optical transmission is a key channel in human-machine interaction, and tactile sensing is essential for enabling a robot human-like perception. The manuscript reported a self-

powered optical tactile sensor inspired by dinoflagellate bioluminescence for real-time visualized tactile perception. By utilizing a self-fabricated pressure sensor as the tactile recognition unit, an electroluminescent component for optical emission and a triboelectric nanogenerator for power supply, the tactile stimuli can be transduced into corresponding optical intensity in a self-powered way. Their device also demonstrated record-high luminescent sensitivity, record-low detection limit in mechano-optical conversion with ultrawide working range. In addition, a robotic hand with high-sensitivity visual tactile recognition capabilities has been demonstrated, highlighting the promising application prospects of their sensor. Therefore, I highly recommend this article for publication in Nature Communications once the following questions are addressed:

Response: Thanks for your very positive comments.

Q1: The sensor's response time to pressure is also important for assessing its mechanical performance. Could the authors provide a test to evaluate the dynamic response speed of the capacitive pressure sensor?

Response: We sincerely appreciate your highly valuable suggestion. Pressure-induced mechanical response constitutes an essential performance characteristic in sensor evaluation. To evaluate the dynamic response characteristics of our capacitive pressure sensor (CPS), a 50 kPa load was applied to it followed by rapid release, revealing a response time of 67 ms and a relaxation time of 80 ms, as demonstrated in Supplementary Fig. 13 (Page 14, Supplementary Materials). This temporal performance is comparable to human skin (~50 ms), implying the exceptional response speed of our CPS to mechanical stimuli.

Modification: Line 160-163, Page 8

To evaluate the dynamic response of the sensor, a 50 kPa load was applied to it followed by rapid release, revealing a response time of 67 ms and a relaxation time of 80 ms

(Supplementary Fig. 13), which are comparable to human skin (~ 50 ms).

Supplementary Fig. 13. Dynamic response time of the CPS under a load of 50 kPa.

Q2: In the section on “Design and sensing properties of the CPS”, it is mentioned that a signal-to-noise ratio of 3:1 can be defined as the detection limit of the sensor. Please provide a reference to support this statement.

Response: Thank you for pointing this out. The references (*Adv. Mater.*, 2023, 35(29): 2300855; *Anal. Chem.* 2019, 91, 1222-1226) have been added to the manuscript to support our statement on the signal-to-noise ratio of 3:1.

Modification: Line 156-157, Page 8

Moreover, a signal-to-noise of 3:1 can be defined as the detection limit of the sensor^{32, 33}.

Q3: Please provide a CIE chromaticity diagram of the emitted light.

Response: We appreciate the reviewer’s suggestion. The corresponding CIE diagram of the emitted light has been supplemented, as provided in Supplementary Fig. 14 (Page 15, Supplementary Materials).

Supplementary Fig. 14. The CIE chromaticity diagram of the emitted light.

Q4: In Fig. 5C, the authors have demonstrated the capability for visual tactile sensing under ambient light through relative brightness between a working ACEL and a reference ACEL. Could the authors also provide a visual comparison under dark conditions?

Response: In response to the reviewer’s insightful advice, Supplementary Fig. 11 (Page 12, Supplementary Materials) now provides a systematic visual comparison of a working ACEL and a reference ACEL under varied loading pressures in dark conditions, revealing distinct optical differentiation. Based on the experimental evidence presented in Fig. 5c, it can be conclusively demonstrated that our SOTS exhibits robust application potential across all environmental conditions.

Modification: Line 342-345, Page 16-17

Additionally, Supplementary Fig. 11 shows the luminance variations of the working and reference ACEL units captured in the darkroom, confirming the robust luminescent distinguishability of the SOTS across all tested environmental conditions.

Supplementary Fig. 11. The luminance variations **a** and their relative intensity profiles **b** of the working and reference ACEL units under varied loaded pressures in darkroom.

Q5: In Fig. S2, the output voltage of R-TENG is tested by introducing a 5 G Ω resistor. Could the author also provide the corresponding output current?

Response: Thanks for the reviewer's suggestion. The corresponding current outputs generated at different rotator heights are supplemented in Supplementary Fig. 3d (Page 4, Supplementary Materials).

Supplementary Fig. 3. R-TENG serves as a high-voltage AC power source within the STOS system. **a** The structure and materials of the R-TENG allow for voltage output regulation by adjusting the upper rotator. **b** Working principles of R-TENG based on triboelectric effect and electrostatic induction. **c** The measurement principle of voltage output of R-TENG. **d** Current output generated by different rotator heights. **e** The corresponding voltage output generated by different rotator heights.